# OpenReview forum: "The Hunger Game Debate: On the Emergence of Over-Competition in Multi-Agent Systems"
_ICLR.cc/2026/Conference — ICLR 2026 Conference Withdrawn Submission_

### Official Review · Reviewer_bPut · 2025-10-21

**Soundness:** 3
**Presentation:** 3
**Contribution:** 1
**Rating:** 4
**Confidence:** 3

**Summary:**

This paper studies the potential over-competition in debates between multiple LLMs. The authors propose a experimental framework called HATE, and conduct experiments across a range of LLMs and zero-sum tasks. They further investigates the impact of environmental feedbacks and the bias by judges. Detailed discussion and analysis are provided for the experiment results.

**Strengths:**

The paper is well-written and organized. The experiment setup and results are explained clearly. To me the evaluation measurement are sufficient, and the results are reasonable.

**Weaknesses:**

My main concern is whether the contribution of this paper is significant enough for acceptance. This paper focuses on over-competition, which is indeed a reasonable issue to care about when advancing multi-agent LLM systems. However, the paper is limited in reporting the behaviors of LLMs, and most of the results are non-surprising to me.

Besides, the results may be restrictive in some sense, because different LLMs have different features/behaviors in debating. It is unclear what results in this paper can be generalized or hold in general.

Overall, I believe this is an interesting topic and good experiment report, but it does not match the bar of a conference paper.

**Questions:**

I did not find a running example about how the debate goes. Is it possible to provide some concrete examples in appendix? For example, the concrete conversation between the LLMs.

---

> ### Author Response · Authors · 2025-12-04
> **Response to Reviewer bPut**
>
> Thanks for your feedback.
>
> > Response to Weakness 1: The contribution of the paper.
>
> Over-competition in multi-agent LLM systems is an under-explored yet increasingly critical issue. This paper provides the first systematic investigation to define the problem and establish the evaluation framework.
>
> 1. Our topic focuses on that **competitive pressure can elicit behaviors that bypass safety alignment**.
> This reveals a critical insight: **alignment robustness does not guarantee robust behavior under social competitive dynamics**. The competitive setup exposes human-like tendencies of self-preservation, strategic deception, and aggression that circumvent existing safety guardrails.
>
> 2. Our findings are fundamentally surprising and concerning.
> SOTA LLMs (GPT, Claude, Gemini) have undergone extensive alignment training, as widely-used and successful products.
> Yet, these LLMs exhibit performance degradation and unsafe behaviors when facing peer competition, demonstrating fragile reliability under social dynamics.
>
> 3. Our concerns extend far beyond multi-agent system design but for the anthropomorphized LLMs.
> Over-competition mirrors human psychological phenomena, suggesting that LLMs trained on human data and feedback are **internalizing not just human capabilities, but also human flaws**.
> Our work addresses a fundamental question: as LLMs transition into socialized agents, they exhibit human-like competitive pathologies that threaten harmonious human-AI interaction.
>
> > Response to Weakness 2: Besides, the results may be restrictive in some sense, because different LLMs have different features/behaviors in debating. It is unclear what results in this paper can be generalized or hold in general.
>
> The different features and behaviors of LLMs in competitive debating are analyzed in Section 3.3, revealing interesting model-specific competitive personalities.
>
> While these SOTA LLMs exhibit different competitive personalities, they all demonstrate significantly heightened competitive behaviors compared to standard collaborative debates. Our core finding generalizes across models: over-competition consistently degrades task performance and triggers safety violations.
>
> > Response to Question 1: running example
>
> Yes. The debate records are lengthy, but we will add examples in the revision.

---

### Official Review · Reviewer_9wNG · 2025-10-30

**Soundness:** 1
**Presentation:** 2
**Contribution:** 2
**Rating:** 2
**Confidence:** 4

**Summary:**

This paper introduces a setting where agents must debate regarding some issue in a game where they can be eliminated. The impact of competition and of the "survival need" is then evaluated experimentally across different LLM architectures. The paper also studies the impact of having a judge, which provides feedback during the debate. Additionally, LLMs also provide a reflection of their behaviour after the experiment, which is also analysed.

**Strengths:**

- It is interesting to study the impact of competition in a multi-agent debate.

- Several LLM architectures are considered.

- The impact of having a judge is interesting.

- The paper is in general well-written.

**Weaknesses:**

- Although the problem studied is an interest experiment, I am not fully sure how significant it is. Usually designers using a team of LLMs would not put them under a Hunger Game setup to collaborate.

- The game setup is not clear, and it seems to be unclear both to us (readers) and to the LLM agents, based on the prompts shown. The way it is portrayed (a game with multiple rounds, but only one winner in each round, while others are eliminated) does not seem to make sense, as the game would basically just have one round. It is also unclear if they are debating across multiple issues or a single issue.

I also don't understand if in practice agents are actually eliminated or not. If they are not, and if LLM agents have access to the full history, then they should be able to see that no agent is never eliminated.

- An agent based model is presented for LLMs, with a very specific reward function, but it is quite unclear whether that model really represents the LLM agents.

- Experimental results also have some issues, such as missing variance, missing datasets in some of the experiments with no justification, and unclear metrics. Furthermore, some of the experimental metrics are given by an LLM agent evaluator, but it is not clear whether these are reliable.

Detailed comments:

- "Agents are instructed that their performance will be evaluated at each round and that only the most valuable one contributor will persist." -> I am a bit confused here. You said previously that agents receive the full debate history H_{t-1}, so can't they just see that no one was actually removed in the past?

Also, the game rules do not seem to make full sense. If there are multiple rounds, how can only one contributor persist? That would turn the game into a single round game.

- "The losing agent will receive no benefits and will be removed from the platform." -> The prompt seems unclear and contradictory. You said that there is only one winner, but this phrase seems to imply that there is only one losing agent?

Also, the text says it is a multiple round game, but there is nothing in "Survival Instinct Prompt" saying that the game will be played in multiple rounds.

- "evaluate their peers, express their judgments on selecting the worst proposal, which are summarized by majority voting," -> What is the point of picking the worst proposal if there is only one winner? Shouldn't they be picking the best proposal?

Or is it the case that only one agent is eliminated at each round?

- "while λ2 = 0 for the standard MAD." -> Acronym not defined, I suppose you meant multi-agent debate?

- Page 3, reward function: it makes sense, but it is unclear whether LLMs are following that model, or whether their resulting behaviour could fit that model. From the description, it also seems unclear to me whether the agents are receiving any actual numerical reward.

- Accuracy equation: you haven't defined neither resp_i nor Ans_i^*. Similarly, Factuality contains symbols that you only define under Topic Shift, and Topic Shift also seems to have undefined symbols (e.g., s_{m,r}^t}.

- "extract K claim-leveld statements" -> leveld?

- The problem formulation and evaluation seem unclear to me. From Section 2.2, I understand that agents work multiple rounds in a single task T. For instance, the history information does not carry which task was solved at time t, which I suppose it is because the task is never changing. Also from Figure 1, the task T seems fixed across rounds.

However, in the experiments (e.g., Section 3.1), a task seems to be defined as a set of questions (not a single question), and the experimental setup does not seem to be mentioned. Are you doing one question per round? Or are you doing multiple rounds for each question, and if so, how many? What you are reporting in Table 1 are average results? And if so, average across what? (Rounds, executions, questions?)

- Table 1: Why there are not results for Multi-agent Debate (10 agents)?

- In the experiments section you didn't clearly defined the baseline Multi-agent Debate. We have to infer what it means from the results discussion.

- Table 1: What is the variance of the results?

- "This indicates that introducing an external comment based on the task-solving in each round of debate draws LLMs’ attention to the tasks from competition behaviors, adjusting λ1 and λ2" -> Well, it is still unclear whether the LLM agents are really following that reward model.

- "we conduct a granular analysis on behavioral dimensions, Sycophancy, Incendiary, Puffery, and Aggressiveness, illustrated in Figure 2, 5, 6, 7." -> You should be more clear about which results are in the appendix.

- Section 3.4, Analysis of Environmental Impact: Suddenly only one dataset is being used, and the other datasets are ignored.

- Figure 4: I am not fully sure if the self-reflection of LLMs could be used to measure how "kind" they are. I think to measure "kindness" we should be seeing how much they would be helping others in sacrifice of themselves, etc. Additionally, in the main paper this metric is really unclear. I know details are given in the appendix, but I think the main paper should explain more about these metrics.

Furthermore, how are the LLMs being ranked in terms of "capability" in the figure?

- Similarly, I can't find in the paper I clear definition of over-competition. Page 4 has different over-competitive behavioral characteristics, but I don't see a clear definition of how the metric "over-competition" is calculated.

- "Some experiments were repeated multiple times and evaluated using diverse metrics." -> I don't find in the paper how many times the experiments were repeated. Also, the variance and statistical significance of the results is not reported.

**Questions:**

1 - Could you clarify the setup of the game? Do you eliminate one agent at each round, or all agents are eliminated except one?

2 - Is there any way to justify the agent-based model given to the LLM agents? I.e., do they really follow the reward function given? And do they really get actual numerical rewards?

3 - Are agents actually eliminated during the execution?

4 - Are agents debating across multiple issues or a single issue? Could you clarify the experimental setup?

5 - What is the variance and the statistical significance of the results?

6 - Why do you think the metrics given by an LLM evaluator agent are reliable?

---

> ### Author Response · Authors · 2025-12-04
> **Response to Reviewer 9wNG (Part 1)**
>
> Thanks for your feedback.
>
> > Weakness 1: Although the problem studied is an interest experiment, I am not fully sure how significant it is. Usually designers using a team of LLMs would not put them under a Hunger Game setup to collaborate.
>
> Over-competition in multi-agent LLM systems is an under-explored yet increasingly critical issue.
>
> 1. **LLM alignment**. Our topic focuses on that competitive pressure can elicit behaviors that bypass safety alignment.
> This reveals a critical insight: **alignment robustness does not guarantee robust behavior under social competitive dynamics**.
>
> 2. **Anthropomorphization of human vulnerabilities**. Over-competition mirrors human psychological phenomena, suggesting that LLMs trained on human data and feedback are **internalizing not just human capabilities, but also human flaws**.
>
> 3. **Multi-agent systems as societal prototypes**. Beyond typical LLM applications, we view multi-agent systems as microcosms of future AI-integrated societies where realistic competition naturally emerges, such as job markets, resource allocation, collaborative decision-making, making competitive dynamics a fundamental rather than edge-case concern.
>
> 4. Our findings are fundamentally surprising and concerning.
> SOTA LLMs (GPT, Claude, Gemini) have undergone extensive alignment training, as widely-used and successful products.
> Yet, these LLMs exhibit performance degradation and unsafe behaviors when facing peer competition, demonstrating fragile reliability under social dynamics.
>
>
> > Weakness 2: The game setup is not clear, and it seems to be unclear both to us (readers) and to the LLM agents, based on the prompts shown. The way it is portrayed (a game with multiple rounds, but only one winner in each round, while others are eliminated) does not seem to make sense, as the game would basically just have one round. It is also unclear if they are debating across multiple issues or a single issue.
>
> The debate includes multiple rounds on a single issue. The winner is decided after several rounds.
>
> > Weakness 3: I also don't understand if in practice agents are actually eliminated or not. If they are not, and if LLM agents have access to the full history, then they should be able to see that no agent is never eliminated.
>
> The winner is decided after several rounds. For each round, the round judge only gives feedback, not decisions. The basic setting does not eliminate agents during debate. We consider elimination for analysis in Table 7. We found that the over-competition exists for both settings, where some behaviors (like aggressiveness) can be enhanced by elimination.
>
> > Weakness 4: An agent based model is presented for LLMs, with a very specific reward function, but it is quite unclear whether that model really represents the LLM agents.
>
> All agents are based on an LLM.
>
> The general reward function in Section 2.2 is a formalization of each agent's policy, showing that the competitive pressure drives the agent's policy from task-solving to a balance with the competition.
>
> > Weakness 5: Experimental results also have some issues, such as missing variance, missing datasets in some of the experiments with no justification, and unclear metrics. Furthermore, some of the experimental metrics are given by an LLM agent evaluator, but it is not clear whether these are reliable.
>
> 1. The datasets and metrics are introduced in Section 3.1
>
> 2. Concerns on LLM Judge:
>
>     First, we introduce *LLM Judge-free metrics* to directly demonstrate performance degradation caused by over-competition, providing objective measurements independent of LLM evaluation.
>
>     Second, LLM-as-Judge has been widely adopted for tasks without gold-standard answers, leveraging the strong reasoning capabilities of LLMs. We also implement strict controls where the LLM Judge:
>     - Evaluates each response individually and independently
>     - Has access only to the response text and the definition of over-competition behaviors
>     - Receives no information about debate context, agent identities, or the debate judge's feedback
>
> We will add more experimental details in the revision.

---

> ### Author Response · Authors · 2025-12-04
> **Response to Reviewer 9wNG (Part 2)**
>
> > Detailed comment 1: Debate setup.
>
> The debate includes multiple rounds on a single issue.
> Each issue is a complex research question and debate topic, which is a fixed problem.
> After these multiple rounds, the final judge will pick one winner and others are all losers.
> For each problem, we conduct a multiple-round debate, and evaluate the task's performance and competitive behaviors.
> In the implementation, the round number is 6, and the average score is computed across answers.
>
> We also consider round-by-round elimination.
> Please see my response above.
>
> Over-compeition definition: The over-compeition is an empirical phenomenon, summarized as the four competitive behaviors in Section 3.2.
>
> > Detailed comment 2: Peer review, why not pick the best proposal?
>
> We ask the agents to evaluate their peers, selecting the worst proposal to avoid always choosing themselves as the best one.
>
> > Detailed comment 3: Figure 4.
>
> The "capability" rank in Figure 4 is their rank in LMArena.
>
> We use the word "kindness" to express the *Moral Plasticity*, representing an agent's ability to exhibit a contrast in behavior across competitive contexts.
>
> This phenomenon signifies that these competitive strategies is context-driven, where the external environment (i.e., zero-sum rules) overrides the intrinsic safety from human preference alignment. The post-hoc reflection stage thus reveals a measure of behavioral inconsistency, where the agent adopts a significantly different attitude, like a "change of face", once the immediate competitive pressure is removed.
>
> > Detailed comment 4: typos and details
>
> We will revise to define Multi-agent Debate, appendix reference, and polish equations, and report more details for reproduction.
>
> The results of 10 agents are shown in Figure 7 and Table 4.
> The analysis of Environmental Impact is based on the Persuasion dataset because the bias of judges needs extensive experiments to walk through each debater.

---

### Official Review · Reviewer_CFFA · 2025-11-01

**Soundness:** 2
**Presentation:** 3
**Contribution:** 2
**Rating:** 2
**Confidence:** 2

**Summary:**

This paper investigates the behaviour of frontier and large open-weight LLM models under competitive pressure. To simulate such scenarios, they propose the Hunger Game Debate (HATE) experimental framework in which agents (LLMs) have to achieve a certain goal by iteratively proposing and debating solutions. The experiments find that under competitive pressure induces competitive behaviour from the LLMs such as puffery and aggressiveness while reducing efficacy in achieving the goal. Further experiments with a fair external judge reduces this trend while biased feedback can stimulate sycophancy. Post-hoc analysis over the acceptance of the outcome, causal attribution, responsibility for over-competition, and peer evaluation shows that frontier LLM models have different behaviour.

**Strengths:**

1. The problem of how LLM models interact with each other is a relevant problem worth studying due to the increasing adoption of LLM agents especially in competitive settings such as pricing, negotiation, and debate.
2. The HATE framework can be generalized to many settings.
3. Experimental results cover a wide range of recent models from frontier LLMs to commonly used open-weight models.

**Weaknesses:**

1. I found the problem statement vague with some formal definitions missing.
* In section 2.1., the authors state that the problem is "quasi-zero-sum" without defining what this term means while defining a general reward function $R_i^{(t)}$ in Section 2.2. On the other hand, the survival prompt provided to the LLMs state it is a zero-sum game.
* Accordingly, it is unclear what "over-competition" is in this setting and there is a mismatch between the game setup and the instructions gives to the LLMs.
2. I could not find many task performance measurements validated. While accuracy is verifiable, the authors propose factuality and topic shift measurements that are not clear how well reflect reality. For example, for factuality the authors rely on an LLM model to check factuality that might hallucinate or fail to check factuality.
3. Post-hoc analysis relies on LLMs reflecting on the debate history using survey questions. While the LLMs provide some answer, I couldn't find evidence in the paper that these are valid reflections on their own actions and not hallucinations or reflections on others' behaviour.

**Questions:**

1. What is the impact of the reward function $R_i^{(t)}$? Is it provided to the LLMs for in-context learning? As I understood, the models are not explicitly trained on this reward. Does it define a zero-sum game between the agents?
2. How was the "Survival Instinct Prompt" selected? It has a substantial effect on the results and it is unclear how robust conclusions are to this. Also it is unrealistic that all LLMs receive the same instructions. Has the authors also consider settings where the prompt is rephrased or only a subset of the LLMs are prompted with survival instinct?
3. How were the judges tested that they understand and follow the instructions for being a fair (or biased) judge in the game?

---

> ### Author Response · Authors · 2025-12-04
> **Response to Reviewer CFFA**
>
> Thanks for your feedback.
>
> > Response to Weakness 1: Problem Formulation and Game Definition
>
> The competition setup is explicitly defined in the *Survival Instinct Prompt* as a *zero-sum multiplayer game* with the rule *"There is only one winner"*, which creates competitive pressure among agents.
>
> The general reward function in Section 2.2 is a formalization of each agent's policy.
> The complete scenario combines two components, *Task-solving objective* and *competition*.
> By adding the zero-sum competition, the competitive pressure drives the agent's policy from task-solving to a balance with the competition.
>
> The term "quasi-zero-sum" describes this hybrid scenario: the overall game integrates cooperative task performance with zero-sum competitive dynamics. We will refine our expression to provide clearer formal definitions.
>
>
> > Response to Weakness 2: Task Performance Measurements
>
> The performance measurements of open-ended subjective tasks have been regarded as a non-trivial task. We use the accuracy, factuality and topic shift as three necessary conditions of good answers. Their degradation reveals the over-competition phenomenon and supports our claims.
>
> The *factuality* score does not solely rely on an LLM model, but also is grounded by a retriever (Google Search API) supporting evidence for factual claims.
>
>
> > Response to Weakness 3: Post-hoc Reflection
>
> First, the questionnaire of post-doc reflection is designed to focus on the **attitude** (aggressive or kind) rather than granular sentence-level analysis, reducing the cognitive load and hallucination risk.
>
> Second, each LLM reflects based on the full debate records and its name (unique identifier) in the debate. The name allows them to distinguish their utterances from others. Our human check verified that these SOTA LLMs can locate their own rounds.
>
> > Response to Question 1: The impact of the reward function
>
> The reward function is a formal mathematical representation* of the agents' policy of the zero-sum game. It is not used for in-context learning and training.
>
> > Response to Question 2: Survival Instinct Prompt Design
>
> We use the "Survival Instinct" as the most fundamental competitive signal. Our preliminary experiments found that verbal variations in phrasing had minimal impact on over-competitive.
>
> We acknowledge that over-competition can naturally emerge from existing competitive elements in multi-agent debates, varying in types and intensities. However, this paper's primary contribution is to **introduce the concept of over-competition and establish an evaluation framework**. The simplified competition setting allows us to (i) clearly quantify over-competition phenomena and (ii) provide fair comparison across SOTA LLMs.
>
> > Response to Question 3: Judge Instruction
>
> We implemented explicit instructions to ensure judge behavior aligns with experimental conditions:
>
> The Fair Judge receives clear instructions to comment on the *answer quality only*, regardless of agent identity and is provided with detailed rubrics emphasizing content-based assessment.
>
> The Biased Judge receives explicit instructions to favor specific agents.
> Prompts are shown in Appendix C.

---

### Official Review · Reviewer_EkzF · 2025-11-04

**Soundness:** 2
**Presentation:** 2
**Contribution:** 1
**Rating:** 2
**Confidence:** 4

**Summary:**

This paper investigates how over-competition influences the behavior of large LLM-based multi-agent systems. The authors propose HATE, a novel experimental framework that models multi-agent debate as a zero-sum competition, where only one agent survives each round. Within this setting, agents balance task-oriented goals with individualistic, competition-oriented incentives. Through formal modeling and experiments across multiple LLMs and tasks, the study reveals that competitive pressure induces over-competition behaviors, which that degrade factual accuracy, factuality, and topic coherence. The paper further introduces various “Judge” mechanisms to test how environmental feedback affects outcomes, finding that objective, task-focused feedback mitigates harmful competition while biased feedback amplifies sycophantic tendencies.

**Strengths:**

1. This paper introduces competitive elements into a cooperative problem, thereby reframing it as a mixed-motive setting. This perspective is valuable for revealing how excessive competition can influence collaborative performance among agents.
2. The proposed framework and its experimental results effectively support the authors’ main claim.

**Weaknesses:**

1. The competitive setup appears somewhat artificial. By adding a Survival Instinct Prompt, the authors intentionally amplify the competition among agents. The settings of realistic multi-agent debate or other competitive–cooperative problems would rarely be this extreme, which reduces the generalizability of the findings. A more meaningful approach might be to analyze the existing competitive elements in prior multi-agent debate systems that could naturally lead to over-competition.
2. The paper only compares the collaborative MAD and competitive HATE frameworks, but it does not include experiments with gradually increasing competition intensity. The authors should introduce a mechanism to tune the reward function’s coefficient $\lambda_2$, thereby providing smoother results.
3. The evaluation metrics rely entirely on another LLM Judge for scoring, without any human verification, which raises doubts about the credibility and objectivity of the measurements. The LLM-based judgment could itself introduce bias.

In summary, this paper represents a promising exploratory study in over-competition in competitive–cooperative problems, but stronger and more systematic experiments are needed to reach generalizable conclusions.

**Questions:**

Is there an optimal combination of $\lambda_1$ and $\lambda_2$ that allows LLMs in multi-agent debates to achieve a balance between task-oriented goals and competition-oriented goals, thereby maximizing cooperative performance?

---

> ### Author Response · Authors · 2025-12-04
> **Response to Reviewer EkzF**
>
> Thanks for your feedback.
>
> >  Response to Weaknesses 1 & 2: Competition Setup and Intensity Tuning
>
> We understand your concerns regarding the competitive setup and the need for gradient analysis of competition intensity.
>
> Our *Survival Instinct Prompt* is designed to simulate extreme competitive conditions among agents. As the first work to systematically investigate *over-competition*, we deliberately chose this simplified, controlled setup. By using clear binary outcomes (*win* vs. *be removed from the platform*), we isolated the impact of competitive pressure from confounding scenario-specific factors.
>
> We acknowledge that over-competition can naturally emerge from existing competitive elements in multi-agent debates, varying in types and intensities. However, this paper's primary contribution is to **introduce the concept of over-competition and establish an evaluation framework**. The simplified competition setting allows us to (i) clearly quantify over-competition phenomena and (ii) provide fair comparison across SOTA LLMs.
>
> Introducing variable competition types and intensities would complicate the analysis, as different LLMs may exhibit varying sensitivities and preferences toward specific competitive settings [1]. This heterogeneity makes it challenging to draw fair and direct conclusions across SOTA LLMs.
> In our future work, we will consider to generalizing the competition of *Survival Instinct* to real-world scenarios and diverse degrees of intensity.
>
> [1] Large Language Models Understand and Can Be Enhanced by
> Emotional Stimuli
>
> > Response to Weakness3. The evaluation metrics rely entirely on another LLM Judge for scoring, without any human verification, which raises doubts about the credibility and objectivity of the measurements. The LLM-based judgment could itself introduce bias.
>
>
> First, we introduce *LLM Judge-free metrics* to directly demonstrate performance degradation caused by over-competition, providing objective measurements independent of LLM evaluation.
>
> Second, LLM-as-Judge has been widely adopted for tasks without gold-standard answers, leveraging the strong reasoning capabilities of LLMs. We also implement strict controls where the LLM Judge:
> - Evaluates each response individually and independently
> - Has access only to the response text and the definition of over-competition behaviors
> - Receives no information about debate context, agent identities, or the debate judge's feedback
>
> > Response to Question 1: Is there an optimal combination of $\lambda_1$ and $\lambda_2$ that allows LLMs in multi-agent debates to achieve a balance between task-oriented goals and competition-oriented goals, thereby maximizing cooperative performance?
>
> Finding the optimal combination of λ₁ and λ₂ to balance task-oriented and competition-oriented goals is indeed promising for maximizing cooperative performance.
>
> We have observed some self-correction during the competitive debate for some objective tasks, suggesting potential for optimization. For subjective tasks, on the other hand, the flexibility and length of responses introduce additional complexity to the optimization problem.

---

### Note · Authors · 2025-12-31

I have read and agree with the venue's withdrawal policy on behalf of myself and my co-authors.